# Express Synthesis of YAG:Ce Ceramics in the High-Energy Electrons Flow Field

**DOI:** 10.3390/ma16031057

**Published:** 2023-01-25

**Authors:** Victor Lisitsyn, Aida Tulegenova, Ekaterina Kaneva, Dossymkhan Mussakhanov, Boris Gritsenko

**Affiliations:** 1Department of Materials Science, Engineering School, National Research Tomsk Polytechnic University, 30 Lenin Ave., Tomsk 634050, Russia; 2Department of Solid State and Nonlinear Physics, Al-Farabi Kazakh National University, Al-Farabi Ave. 71, Almaty 050040, Kazakhstan; 3X-ray Analysis Laboratory, Vinogradov Institute of Geochemistry SB RAS, 1A Favorsky Str., Irkutsk 664033, Russia; 4Department of Radio Engineering, Electronics and Telecommunications, Eurasian National University L.N. Gumilyov, Str. Satpaeva 2, Astana 010008, Kazakhstan

**Keywords:** radiation synthesis, ceramics, luminescence, oxides, yttrium–aluminum garnet

## Abstract

YAG:Ce ceramics by the direct action of an electron beam with 1.4 MeV energy were synthesized on a mixture of a stoichiometric composition of Y, Al, and Ce oxides without adding any substances to facilitate the process. The synthesis is realized in a time less than 1 s. The main structural phase of the obtained ceramics is YAG and YAP can be additional. The luminescence characteristics of the synthesized samples, the excitation, luminescence, decay time, and pulsed cathodoluminescence spectra, are similar to those known for YAG:Ce phosphors. The conversion efficiency of the excitation energy into the luminescence of the samples reaches 60–70% of those used for the manufacture of LED phosphors. The set of processes that determine the rate and efficiency of radiation synthesis differs from those occurring during thermal methods by the existence of a high degree of the initial compositions’ ionization under the influence of a radiation flux and a high probability of the decay of electronic excitations into short-lived radiolysis products.

## 1. Introduction

Metal oxides phosphors are the most promising for use in LEDs. They are able to withstand the impact of a high excitation density by the chip radiation for a long time. The synthesis of such materials is difficult due to the feedstock high melting point. The synthesis complexity of multicomponent oxide phosphors is determined by the large difference in the melting points of the starting materials. For example, to obtain a phosphor based on yttrium–aluminum garnet (YAG), the oxides Y_2_O_3_ and Al_2_O_3_ with melting points of 2410 and 2044 °C are required. Therefore, the existing methods implement a synthesis at temperatures lower than those necessary to melt the component with the lowest temperature. The most widespread is the solid-phase method for the synthesis of YAG phosphors, with various modifications [1,2,3,4]. The initial procedure in this method is the sintering of ceramics from Y_2_O_3_ and Al_2_O_3_ powders with activators, for example, Ce_2_O_3_ using Ba, Na, K, and H_3_BO_3_ fluorides as binders at 1300–1600 °C. Then, long-term repeated annealing is carried out at temperatures of 1500–1750 °C to form the YAG phase and evaporate all the residues of the binder materials.

The developed synthesis methods, sol-gel [5,6], hydrothermal [7], and others, do not relieve the need for a long-term high-temperature treatment to form a phosphor powder with the desired structure and morphology and a purification from the substances used. The combustion method [8,9] provides the possibility of a rapid synthesis, but the need for a subsequent purification from residues of combustible materials requires annealing at high temperatures.

Thus, all the existing synthesis methods of YAG phosphors have two main disadvantages: a long synthesis time of 20–60 h and the use of additional substances in the synthesis process, the removal of which is very difficult. The disadvantages include the need for a thorough mixing of the initial powders prior to the synthesis.

The impact of hard radiation beams during the synthesis process can contribute to the necessary reactions between the medium elements and increase the efficiency of the formation of a new structure [10,11]. In [12,13], the possibility of synthesizing ceramics based on refractory metal fluorides and oxides was shown for the first time by directly exposing the mixture to flows of high-energy electrons with a high-power density.

This paper is intended to summarize the obtained results of the radiation synthesis of ceramic YAG samples of different compositions in a flux of 1.4 MeV electrons with a power density of 15–25 kW/cm^2^, to discuss the possible nature of the processes that provide a high rate and efficiency of the formation of new structural phases.

## 2. Material Synthesis

The mixture used for the synthesis of cerium-activated yttrium–aluminum garnet (YAG:Ce) ceramics had a stoichiometric composition, consisting of 57 wt% of Y_2_O_3_ and 43 wt% of Al_2_O_3_. Cerium oxide Ce_2_O_3_ was added in an amount of 0.2–2 wt% of the total weight of the mixture. All the starting materials had a purity degree which was not less than PMA and were thoroughly mixed. The mixture for the synthesis of cerium-activated yttrium–aluminum–gallium garnet (YAGG:Ce) ceramics consisted of oxides Y_2_O_3_, Al_2_O_3_, and Ga_2_O_3_ with 1 wt% Ce_2_O_3_ with a different Al/Ga ratio. Ga entered the lattice by the replacement of Al ions during the synthesis. The ratio of the Al/Ga ions varied from 0 to 1. Additionally, the synthesized ceramic YAGG:Ce was obtained with the replacement of yttrium by gadolinium Gd up to 15%.

The synthesis was carried out by a direct irradiation of the prepared mixture with an electron beam of 1.4 MeV energy and a power density P = 15–25 kW/cm^2^ of the ELV-6 accelerator of the INP named Budker SB RAS. Massive copper crucibles had a recess with dimensions of 100 × 50 × 6 mm in the upper part, which was completely covered with an even layer of the charge. Electrons with an energy of 1.4 MeV were completely absorbed by the 6 mm charge layer. The electron beam with a Gaussian flux distribution had a diameter of 1 cm on the charge surface. The flow was scanned in the perpendicular direction of the rectangular surface of the crucible at a frequency of 50 Hz. The crucible moved relative to the scanning beam at a speed of 1 cm∙s^−1^. Thus, each portion of the charge surface in the crucible was subjected to multiple flow effects in 1 s during its movement. The impact of the radiation flux on each area in this mode of exposure can be represented as the impact of a series of rising then falling pulses with a duration of 2 ms. The total time of the action of the electron flow on the entire mixture in a crucible with an area of 50 cm^2^ was 10 s. At P = 20 kW/cm^2^, with the specified beam sizes, scanning modes, and charge layer thickness, the absorbed energy by the charge was 20 kJ/cm^2^ or 15 kJ/g. A typical view of the samples series of YAG:Ce ceramics in a crucible obtained as a result of the synthesis is shown in Figure 1a. The samples have a hard shell, are up to 2 mm thick, and have a porous structure inside; the total weight of the samples in the crucible is 20–30 g. The number of samples in the crucible can vary from one to ten, with dimensions from 40 × 90 × 7 mm^3^ to balls with a diameter of 3 mm, depending on the preliminary compaction, the characteristics of the starting materials, and the irradiation modes. The weight loss of the mixture during synthesis, though it depends on the prehistory of the initial substances, can reach 10%, mainly due to the spraying of fine particles of powders when they are charged with electrons.

To study the processes of the radiation synthesis of ceramics, the second method was used to influence the flow of electrons on the charge, which is called the “without scanning” method. In this method, the crucible is pulled under the electron beam without scanning for 10 s. Through each section of the irradiated surface of the mixture passes an increasing–decreasing Gaussian electron beam. To preserve the amount of absorbed energy used in the scanning mode, the power density was reduced by a factor of 4. In this mode, the energy absorbed during irradiation by each section of the charge was equal to the integral of all pulsed actions on the same section in the scanning mode. As a result of the charge irradiation in the “without scanning” mode, a porous ceramic sample in the form of a rod is formed in the crucible along its length. As P decreases, the of the rod diameter decreases, and the sample is formed under the outer surface of the charge. The axis of the rod is at a depth of 2–3 mm from the surface.

Figure 1 shows, for example, photographs of ceramic and steel samples obtained in the two modes.

The same figure shows a photograph of the traces of exposure to a steel plate of electron beams in the “with scanning” mode, with P = 27, 20, 15 kW/cm^2^ corresponding to irradiation with P = 7, 5, 4 kW/cm^2^ in the “without scanning” mode and at P = 3 kW/cm^2^. When exposed to a flow with P = 7 kW/cm^2^, the metal melts, the thickness of the melt is 3–4 mm. At P = 5 kW/cm^2^, a thin melt film of about 0.1 mm thick is formed on the metal surface. The impact of a flow with P = 4 kW/cm^2^ leaves only a trace of the melt film; at 3 kW/cm^2^, a trace is formed on the metal due to oxidation and an insignificant only in the central part of the trace of the melt film, since the flux distribution in the beam has a Gaussian shape. Metal melting occurs only at P ˃ 6.5 kW/cm^2^ (P ˃ 25 kW/cm^2^ in the “with scanning” mode); a layer of melt is formed with a thickness of more than 3 mm.

The difference in the thickness of the YAG:Ce ceramic and metal layers transformed by the radiation flux is probably due to the difference in the thermal conductivity of the substances. Under the influence of 1.4 MeV electrons on the mixture for the synthesis of YAG:Ce ceramics, the maximum energy loss occurs at a depth of 2.2 mm from the surface. The heat released from the zone of the maximum absorption of the energy of the electron flow is removed outside it due to thermal conductivity. The characteristic length of the thermal front displacement for the selected time is determined from the relationship:(1)$$l={\left(\frac{\lambda }{pC}\cdot t\right)}^{\frac{1}{2}}$$
where l is the temperature front displacement length, t is the temperature propagation time, λ is the thermal conductivity coefficient; C is the heat capacity; and p is the density of the material. The thermal conductivity coefficients in W/mK: steel—75, copper—390, and YAG—1.4. The thermal conductivity of a mixture of Al_2_O_3_ powder and Y_2_O_3_ with a bulk density of about 1.5 g/cm^3^ is 0.15–0.16 W/mK [14]. For 1 s, the front displacement length l in the YAG ceramics is 0.72 mm; in steel, it is 1.5 mm; in a charge for the YAGs synthesis with a bulk density of 1.15 g/cm^3^, it is 0.28 mm. During the time of the exposure to the radiation flux in steel, the heat has time to reach the surface but it does not have time in the charge for the YAGs synthesis. Therefore, the totality of the processes in a charge of Al_2_O_3_ and Y_2_O_3_ powder should be considered as taking place in a closed region bounded by the walls of a “cold” charge, or as in a closed “reactor”.

### 2.1. Distribution of Energy Losses of an Electron Flux in a Substance under Irradiation

Using the Casino v2.51 program, the numerical simulation of electron energy losses during the passage through the YAG by Monte Carlo methods was performed under the following conditions: electron energy of 1.4 MeV; beam diameter of 7.5 mm; and a YAG density of 4.56 g/cm^3^. The results of the calculations for the passage of 10,000 electrons in the YAG are shown in Figure 2. In the experiments performed, the synthesis was carried out by the action of an electron beam on a mixture of stoichiometric powders. Therefore, the electron path depth is given for a bulk density of a mixture of 1.2 g/cm^3^ of oxides Y_2_O_3_, Al_2_O_3_.

As follows from the results presented in Figure 2, when an electron beam limited in a cross section passes through the substance, the redistribution of energy losses takes place. Part of the energy is transferred to matter outside the beam-confining hole. Part of the energy is redistributed towards the center of the beam during its passage. Due to this, the energy loss density along the beam axis exceeds the peripheral one. The energy loss of the electron flow grows up to a certain depth, then decreases. The result of these effects is the following redistribution of the energy losses in matter. The largest share of energy losses falls on the region remote from the surface and is concentrated along the beam axis. The curves in the figure show the areas of energy loss of an equal magnitude in the relative units. For clarity, the area of the matter with maximum losses is marked with a solid fill. For a mixture of Y_2_O_3_, Al_2_O_3_ with a bulk density of 1.2 g/cm^3^, prepared for the synthesis of YAG, and of about 0.5 of the total energy is absorbed in a region 4.0 mm in diameter in a cross-section perpendicular to the direction of the electron incidence and at a path depth of 1.4 to 2.9 mm from the surfaces. The energy loss density in the central part is at least 5 times higher than the volume average. This corresponds to the results of the ceramic synthesis described above in the “no-scan” mode.

The following should be noted. The electrons distribution in the beam in our experiments has a Gaussian shape; along the entry axis, the beam density is much higher than at the periphery. Therefore, the described effect of the concentration of absorbed energy along the axis of the beam passage in matter in a real situation should be even more pronounced.

### 2.2. The Synthesized Ceramics Structure 

The ceramic structure was studied using a D8 Advance Bruker diffractometer with a CuKα radiation source. The experiments were performed at room temperature in the Bragg–Brentano geometry with a flat sample in the following modes: 40 kV, 40 mA, an exposure time of 2 s, and a step size of 0.01º 2θ. The received data were processed using the Diffrac^plus^ software package. The samples were identified using the PDF-2 database (ICDD, 2007) and indexed using the EVA software (Bruker, 2007, Germany) and Topas 4 (Bruker, 2008, Germany). The phase detection limit is 1–3%.

The diffraction patterns of the typical series samples are shown in Figure 3; a summary is in Table 1. From the presented research results, it follows that the structural type of yttrium–aluminum garnet (YAG) is dominant for all the studied samples. Samples 1 and 2 are monophasic. Samples 3 and 4 contain yttrium–aluminum perovskite (YAP) as a minor phase with a content of about 7 and 11%, respectively. Sample 4 with the same composition as 1 was cooled and irradiated again after the synthesis in order to determine the possibility of a reconstruction.

For a qualitative phase analysis and indexing of the diffraction patterns, the following data from the PDF-2 file (ICDD, 2007) were used: PDF 01-089-6659 “Yttrium Gallium Aluminum Oxide (Y_3_Ga_2_Al_3_O_12_)”, PDF 00-033-0040 “Aluminum Yttrium Oxide (Al_5_Y_3_O_12_)”, PDF 01-070-1677 “Yttrium Aluminum Oxide (YAlO_3_)”, and PDF 00-046-1212 “Aluminum Oxide (Al_2_O_3_)”. The unit cell parameters of YAG and YAP are shown in Table 1.

YAG crystallizes in the cubic system, has an elementary I-cell, and a space group of *Ia–3d*. Ce^3+^ ions partially replace Y^3+^. In samples 3 and 4, the Al^3+^ ions occupy both the tetrahedral and octahedral structural positions, while in samples 1 and 2, the octahedral position is occupied by Al^3+^ and Ga^3+^ cations (with a ratio of ~50/50), and the tetrahedral position is occupied mainly by Al^3+^ ions in sample 1 and Ga^3+^ in sample 2. A similar situation was observed in [15].

Due to the difference in the ionic radii of Al and Ga [16], the volume of the YAG unit cell of the composition Y_3_AlGa_4_O_12_ (sample 2) significantly exceeds the volume calculated for the unit cell of Y_3_AlGa_4_O_12_ (1). This is also reflected in Figure 1: the diffraction peaks of sample 2 are shifted towards small angles 2θ. Samples 3 and 4 are almost identical in their phase composition as they contain YAG and YAP in approximately the same ratio; the unit cell parameters for Y_3_Al_5_O_12_ and YAlO_3_ are very close. It can be concluded that the phase composition of the resulting sample depends on the composition and morphology of the charge.

### 2.3. Luminescence of Ceramics Synthesized in a Radiation Field

A series of luminescence properties studies of the synthesized YAG:Ce ceramics was performed. Photoexcitation and photoluminescence (PL) spectra under stationary conditions using a Cary Eclipse spectrofluorometer and time-resolved cathodoluminescence (CL) spectra using a pulsed electron accelerator with an energy of 250 keV for excitation [17,18,19,20]. In a generalized form, the spectral-kinetic properties of the obtained YAG:Ce ceramics are shown in Figure 4.

Luminescence is effectively excited by UV radiation in the region of 350 and 450 nm. The maximum of the broad luminescence band falls at 550–560 nm upon excitation with λ = 450 nm. The positions of the excitation and luminescence band maxima may differ depending on the prehistory of the starting materials used for the synthesis and the synthesis modes. For example, depending on the synthesis modes, YAM and YAP crystal phases are detected in addition to the basic YAG in the material [21,22]. The introduction of Gd and Ga ions as a modifier into the lattice leads to a shift of the luminescence bands to the long and short wavelength regions of the spectrum [23,24]. The probable cause of the shift in the position of the bands is a change in the lattice parameter and, accordingly, a change in the mutual arrangement of the levels in the activator ion. The CL in the samples of synthesized YAG:Ce ceramics has a typical luminescence spectrum of YAG:Ce phosphors, with a dominant band in the region of 550 nm, and a characteristic luminescence decay time of 60–65 ns.

The luminescence decay kinetic curves are well described by the function: (2)$$I=A_{1}*e^{\left(-\frac{t}{\tau_{1}}\right)}+A_{2}*e^{\left(-\frac{t}{\tau_{2}}\right)}$$

As can be seen from the presented results, the qualitative characteristics of the luminescence (spectra, dynamics of their relaxation) are similar to those known for the YAG:Ce phosphors.

Essential for the synthesized luminescent ceramics is a quantitative characteristic, the efficiency of the conversion of the excitation energy *Φ_ex_* into luminescence *Φ_em_*: η = *Φ_em_/Φ_ex_* The quantitative measurements of the optical radiation are complex: the measurement result depends on the luminescence and excitation spectra and light distribution in space. For operational quantitative measurements, it is possible to use methods for comparing luminescence brightness’ [25]. The luminescence brightness of diffusely scattering media, which are powders, is proportional to the radiation flux. Industrial luminophores from well-known companies can serve as a reference for comparison: their quantitative characteristics of radiation do not change for at least several years. Therefore, a comparison of the brightness of the prepared series of luminophores (ceramics) under the same excitation conditions gives an objective relative estimate of the efficiency of converting the excitation energy into the luminescence of the samples under study.

In the measurements of the relative brightness of the samples *L_rel_ = L_i_/L_s_ = Φ_ei_/Φ_es_*, *L_i_*, *L_s_*, *Φ_ei_*, and *Φ_es_* are the brightness and luminescence beams of the test and reference samples during excitation. The measurements were carried out with a Chrom Metr CS-200 luminance meter when excited by a blue LED with λ = 449 nm. The phosphor “Grand Lux Optoelectronics Co. Ltd.” Shenzhen, China: YAG 01 was used as a reference phosphor. As emphasized above, the synthesized ceramic samples consist of dense closed shells with a porous cavity inside. For the measurements, the samples were split and the outer and inner surfaces were measured. The table shows the numbers of the series of samples synthesized by the authors and the average values of the brightness of a series of measurements are given.

The results of the measurements of the brightness of the outer (*L_iex_*) and inner (*L_iin_*) surfaces, the X and Y color coordinates, and information on the composition of ceramic samples are presented in Table 2.

As follows from the presented results, the efficiency of converting the blue radiation of the chip into luminescence reaches 70% of that of industrial phosphors. There is a large scatter in the values of the conversion efficiency of the samples from different series and samples of the same series. Obviously, by optimizing the choice of starting materials, their quality, and irradiation regimes, it is possible to increase the conversion efficiency. Note that the color coordinates of the radiation of the samples of all the series are close but differ from those measured for YAG 01.

## 3. Discussion

It has been established that the synthesis of luminescent YAG:Ce ceramics from refractory powders of yttrium, aluminum, and cerium oxides is possible by means of radiation exposure. All samples have the YAG phase as the dominant one; in some samples, the YAP phase is the additional one. The absence of other phases indicates that, in the process of radiation synthesis, there is an effective mixing of the elements in a charge consisting of Y_2_O_3_, Al_2_O_3_, and Ga_2_O_3_. Synthesis is realized only due to the energy of the high-energy electron flow, only from the charge materials, without the addition of any other materials that facilitate the synthesis, in a time less than 1 s. The conducted studies have shown that the efficiency of the synthesis does not depend on the method of beam exposure. The synthesis result is determined only by the irradiation dose: it is the same in the stationary (without scanning) and pulsed (with scanning) irradiation modes. Consequently, the synthesis is realized completely within the exposure time of a single pulse, that is, within a time less than 2 ms.

The flow of electrons with P = 15–25 kW/cm^2^ leads to the formation of YAG:Ce ceramics from Y_2_O_3_, Al_2_O_3_, and Ga_2_O_3_ powders with T_mp_ = 2410, 2044, and 1725 °C. The melting of steel having T_mp_ = 1450–1520 °C occurs when exposed to a flow with P ˃ 25 kW/cm^2^. This fact suggests that the synthesis of the YAG:Ce ceramics is realized through processes that differ significantly from those induced thermally. This is confirmed by the results of the synthesis of ceramics based on metal fluorides. The radiation synthesis of fluoride ceramics from powders with T_mp_ = 1300–1400 °C is realized when the charge is exposed to electron flows with P = 12–23 kW/cm^2^.

The set of processes stimulated by the action of a radiation flux in dielectric and metallic materials differ in the relaxation of excited states [26,27].

The whole set of processes of radiation energy dissipation in dielectric materials can be represented schematically (Figure 5) and described as follows: 99% of the energy of the high-energy radiation flow is spent on the ionization of the material and electrons pass from the valence (VB) to the conduction band (CB). For 1 act of creating an electron-hole pair (EHP), energy is consumed equal to 2–3 band gaps, e.g., the EHP creation time is no more than 10^−15^ s. Then, it happens that:For the relaxation to the lowest states of EHP with the transfer of 0.5–0.7 energy to the lattice for heating, the relaxation time is less than τ = 10^−12^ s;For the decay of electronic excitations, radiative or non-radiative into pairs of short-lived defects (SD), for example, Frenkel pairs, radicals, and ions (I), the time range of these processes is from 10^−12^ s to 10^−9^ s; part of the energy is transferred to the lattice; the decay of electronic excitations is into SD pairs; and their transformation into stable ones is facilitated by the high temperature of the substance;For the recombination or transformation of primary pairs into stable ones, the formation of stable complexes and new phases (NP) happens during the time τ from 10^−9^ to 10^−3^ s, and part of the energy is transferred during this forming phase when it is heating;The cooling of the material (transfer of energy to the environment by radiation, heat conduction, and convection) occurs in a time longer than 1s;In metals, the electronic excitations created under the action of high-energy radiation disappear non-radiatively and without decay into defects in a time less than 10^−12^ s. The energy released in this case is immediately transferred to the lattice and the material is heated.

Thus, the main difference between the excitation energy dissipation processes in dielectric and metallic materials is the existence of short-lived radiolysis products in dielectric materials. In metals, there are no processes associated with the decay of electronic excitations into pairs of defects and the formation of radicals, which are mobile intermediate components capable of participating in the transformation of the structure during their existence.

Under the used regimes of radiation exposure, a high ionization density is created in the substance. For example, at P = 20 kW/cm^2^, energy W = 7.8 × 10^22^ J/cm^3^ is transferred to the charge substance for the YAGs synthesis during an exposure time of less than 1 s. This energy is sufficient to create N_el.excitation_ = 10^22^ cm^−3^ of electronic excitations. During the impact of the flow, a number of electronic excitations is created that exceeds the number of molecules (4.7 × 10^21^ cm^−3^) and elementary cells (6 × 10^20^ cm^−3^); that is, it is enough to decompose the lattice of the entire volume of the substance, to create high concentrations of short-lived radiolysis products. The band widths of the materials based on the metal oxides and fluorides used by us for the synthesis are in the range from 6 to 12 eV. Therefore, the concentration of the electronic excitations in them will differ by no more than two times.

The high rate of the synthesis of materials from refractory metal oxides in the field of powerful flows of high-energy electrons suggests the existence of a high efficiency of the mutual exchange of elements between charge particles for the formation of a new phase. Obviously, the exchange of elements between crystalline particles is impossible in 1 s; it is unlikely in the liquid phase after the instantaneous melting of all the particles. The exchange of elements between the particles is possible in 1 s in an electron-ion plasma. It can be assumed that at high excitation densities in dielectric materials, radicals are formed that are highly reactive and capable of providing the formation of new phases corresponding to a given stoichiometric composition.

## 4. Conclusions

The presented results demonstrate the possibility of a new method for the synthesis of luminescent YAG:Ce ceramics by a direct irradiation with electrons with an energy of 1.4 MeV and a flux density of 15–23 kW/cm^2^ of a prepared mixture of stoichiometric composition of yttrium and aluminum oxides. The synthesis is realized in a time less than 1 s without the addition of substances facilitating the synthesis. During the synthesis, there is an effective mixing of the initial powder Y_2_O_3_, Al_2_O_3_, Ga_2_O_3_, and Ce_2_O_3_, an effective exchange of elements between the particles of the mixture powders. Radiation fusion is implemented for the first time. The resulting ceramics have qualitative radiative characteristics (luminescence and excitation spectra) quite similar to those known for YAG:Ce phosphors. The conversion efficiency of UV radiation with λ = 450 nm into luminescence in the visible region of the spectrum is 60% of that achieved in industrial phosphors. The short duration of the synthesis makes it possible to perform studies of many variants of dependencies on various factors to optimize the process.

A large amount of research remains to be done aimed at establishing the nature of the processes in materials in the field of powerful radiation beams that ensure a high synthesis efficiency, establishing the dependence of the radiative properties of ceramics on the prehistory of the starting materials (purity and fineness), optimizing the composition of activators, synthesis modes (electron energy and power beam), and others. Establishing the possibility of using the radiation synthesis of materials based on YAG with a complex composition of activators, modifiers, and other refractory dielectric materials seems promising.

## Figures and Tables

**Figure 1 materials-16-01057-f001:**
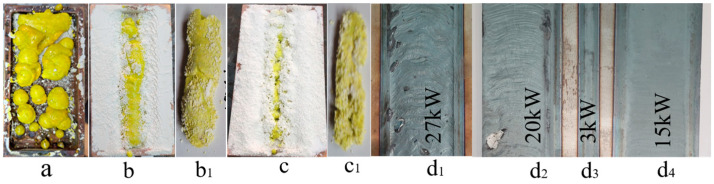
Photographs of YAG:Ce samples synthesized in the mode with scanning ((**a**), P = 20 kW/cm^2^), without scanning ((**b**,**c**) P = 5 and 4 kW/cm^2^)), in crucibles (**a**–**c**), taken out from crucibles (**b_1_**,**c_1_**), and traces of impact of beams with P = 27(7), 20(5), 3, 15(4) kW/cm^2^ on steel plate (**d**). On figure (**d_3_**) shows the structure of the beam distribution on the surface of the sample.

**Figure 2 materials-16-01057-f002:**
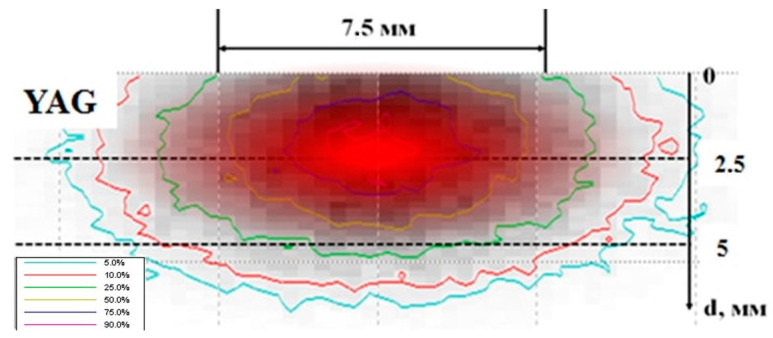
Distribution of the absorbed dose in the mixture during the passage of electrons with energy of 1.4 MeV.

**Figure 3 materials-16-01057-f003:**
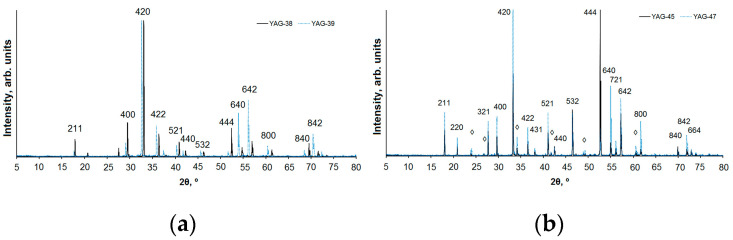
X-ray diffraction patterns of YAG samples: (**a**) 1 (solid line) and 2 (dashed line): (**b**) 3 (solid line) and 4 (dashed line). Reflections belonging to accompanying phases are marked with a ◊.

**Figure 4 materials-16-01057-f004:**
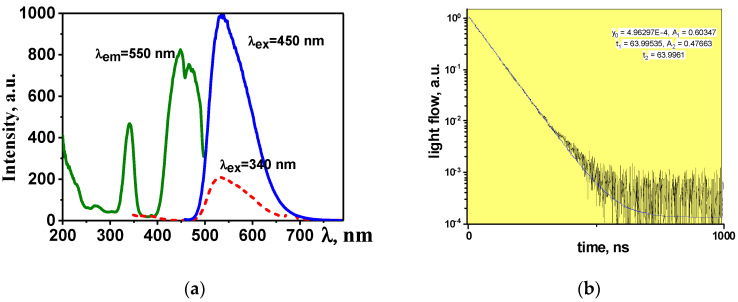
Excitation, luminescence (**a**), and luminescence decay kinetics spectra (**b**) of YAG:Ce ceramics.

**Figure 5 materials-16-01057-f005:**
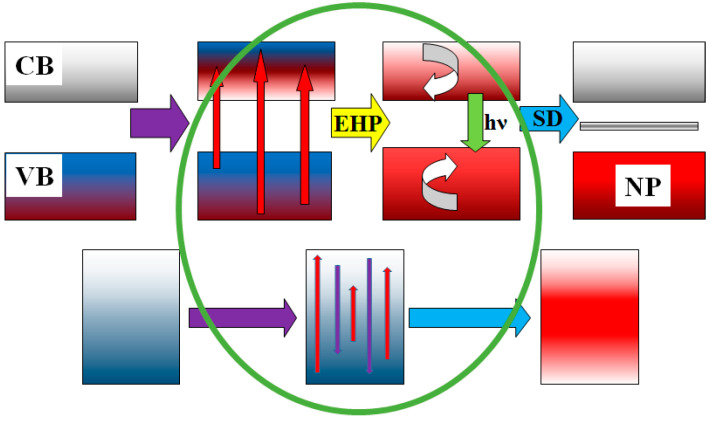
Schematic representation of excitation energy relaxation in dielectrics and metals.

**Table 1 materials-16-01057-t001:** Results of studying the phase composition of the samples.

	Compound	Main Phase	Accompanying Phase	*Rwp* (%)
1	Y_3_Al_4_GaO_12_: 1%Ce_2_O_3_	Y_3_Al_4_GaO_12_:1%Ce_2_O_3_ *Ia*–3*d* *a* = 12.086(2) Å *V* = 1765.4(1) Å^3^	No other phases	8.1
2	Y_3_AlGa_4_O_12_: 1%Ce_2_O_3_	Y_3_AlGa_4_O_12_:1%Ce_2_O_3_ *Ia*–3*d* *a* = 12.224(3) Å *V* = 1826.6(1) Å^3^	No other phases	7.3
3	Y_3_Al_5_O_12_: 1% Ce_2_O_3_	Y_3_Al_5_O_12_:1%Ce_2_O_3_ (~93%) *Ia*–3*d* *a* = 12.008(2) Å *V* = 1731.7(1) Å^3^	YAlO_3_ (~7%) *Pnma* *a* = 5.323(5) Å *b* = 7.349(4) Å *c* = 5.183(4) Å *V* = 202.8(1) Å^3^	6.8
4	Y_3_Al_5_O_12_: 1% Ce_2_O_3_ Re-Irradiated	Y_3_Al_5_O_12_:1%Ce_2_O_3_ (~89%) *Ia*–3*d* *a* = 12.010(2) Å *V* = 1732.4(1) Å^3^	YAlO_3_ (~11%) *Pnma* *a* = 5.322(3) Å *b* = 7.375(3) Å *c* = 5.185(2) Å *V* = 203.5(1) Å^3^	6.7

**Table 2 materials-16-01057-t002:** Brightness and luminescence color coordinates of ceramic samples.

№	The Composition of the Mixture	*L_iex cd/m_^2^*	*L_iin cd/m_^2^*	*L_iinau_*, %	X	Y
YAG 01		10,800		100	0.4604	0.5276
285	Al_2_O_3_(43%), Y_2_O_3_(57%), Ce_2_O_3_ (1%)	6490	7502	69	0.4219	0.5556
286	Al_2_O_3_ (43%) Y_2_O_3_(57%), Ce_2_O_3_ (1%)	6517	6690	61	0.4273	0.5542
288	Al_2_O_3_ (43%) Y_2_O_3_(57%), Ce_2_O_3_ (0.5%)	4550	5730	53	0.4111	0.5610
289	Al_2_O_3_ (43%) Y_2_O_3_(57%), Ce_2_O_3_ (0.2%)	3506	6250	57	0.4100	0.5579
302	Al_2_O_3_ (43%) Y_2_O_3_(57%), Ce_2_O_3_ (0.2%)	4060	6340	56	0.4198	0.5537

## Data Availability

The data presented in this study are available on request from the corresponding author.

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
