# Peer review of "Express Synthesis of YAG:Ce Ceramics in the High-Energy Electrons Flow Field"

_materials, 2023, doi:10.3390/ma16031057_

Round 1

Reviewer 1 Report

Comment on "Express synthesis of YAG: Ce ceramics in the high-energy elec trons flow field" by Lisitsyn et al.

 Metal oxides phosphors are the most promising for use in LED and The synthesis of such materials is difficult due to the feedstock high melting point. Herein Lisitsyn et al. synthesized YAG:Ce ceramics by direct action of an electron beam with 1.4 meV energy on a mixture of stoichiometric composition of Y, Al, Ce oxides without adding any substances to facilitate the process. The synthesis is realized in a time less than 1 s. The main structural phase of the obtained ceramics is YAG, and YAP can be additional. The conversion efficiency of excitation energy into luminescence of the samples are reaches 60-70% of those used for the manufacture of LED phosphors. The set of processes that determine the rate and efficiency of radiation synthesis differs from those occurring during thermal methods. The reported results are useful for functional materials synthesis. However, the manuscript needs improving for clarity, completeness, logic and conciseness. Therefore I suggest the authors revise their manuscript to meet the publication standard of the journal.

The following points are my concerns which are important to improve the clarity, significance and logic of the manuscript. Therefore they should be well-addressed and included in a revised version of the manuscript.

(1) I suggest the authors make a conclusion on the manuscript and write it as Section 5 for clarity and completeness of the manuscript.

(2) In Table, the column for four compounds is blank. Please fill them for completeness.

(3) In second column and fifth line of Table 1, what is the meaning of the phrase "Облучен повторно"?

(4) There is no any description and discussion on Fig. 4 in the main text. What are the meanings of the variables displayed in the inset of Fig. 4b? The authors should give the correlation equation for decay kinetics spectra of YAG:Ce ceramics in the main text with obtained parameters.

(5) In the caption of Figure 2, " with energy 1.4. MeV." should be corrected to "with energy of 1.4 meV"(?). In the caption of Figure 1, "2" in unit "kW/cm2" should be a superscript.

(6) The quality of the YAG:Ce ceramics obtained using the method described in the manuscript should be compared with that obtained using previous methods to demonstrate the advantage of the synthesis method proposed in the manuscript. What are the advantage and disadvantage of the synthesis method proposed in the manuscript?

(7) The format of the references in the Reference List does not match that of Materials. For instances, the first three references should be listed as:

1. Pan, Y.; Wu, M.; Su, Q. Comparative investigation on synthesis and photoluminescence of YAG:Ce phosphor. Materials Science and Engineering: B 2004, 106, 251-256. https://doi.org/10.1016/j.mseb.2003.09.031 333

2. Smet, P.F.; Parmentier, A.B.; Poelman D. Selecting conversion phosphors for white light-emitting diodes. J. Electrochem. Soc. 2011, 158, R37. https://iopscience.iop.org/article/10.1149/1.3568524 335

3. Ye, S.; Xiao F.; Pan, Y.X.; Ma, Y.Y.; Zhang, Q.Y. Phosphors in phosphor-converted white light-emitting diodes: Recent advances in materials, techniques and properties. Materials Science and Engineering: R: Reports. 2010, 71, 1-34. https://doi.org/10.1016/j.mser.2010.07.001.

Author Response

The authors thank you for your detailed analysis and helpful comments. All comments have been corrected:

1) Conclusion done, attached

2) Table 1 corrected. Insertions made: No other phases, Re-irradiated, translation made, clarification given in the text.

3) A description has been added to the text. Sample 4 with the same composition as 1 was cooled and irradiated again after synthesis in order to determine the possibility of restructuring.

4) The formula for the characteristic decay time has been added to the text. The inset in Figure 4b shows the calculation values of the characteristic times.

5) corrected as "with energy of 1.4 MeV"

6) Conclusion attached

7) References corrected, compiled at the request of the journal

We submit the article for translation and proofreading to professional MDPI translators. We hope for a quality translation. 

Conclusion

The presented results demonstrate the possibility of a new method for the synthesis of luminescent YAG:Ce ceramics by direct irradiation with electrons with an energy of 1.4 MeV and a flux density of 15–23 kW/cm2 of a prepared mixture of stoichiometric composition of yttrium and aluminum oxides. The synthesis is realized in a time less than 1 s without the addition of substances facilitating the synthesis, during the synthesis there is an effective mixing of the initial powder Y2O3, Al2O3, Ga2O3 Ce2O3, an effective exchange of elements between the particles of the mixture powders. Radiation fusion is implemented for the first time. The resulting ceramics have qualitative radiative characteristics (luminescence and excitation spectra) quite similar to those known for YAG:Ce phosphors. The conversion efficiency of UV radiation with λ=450 nm into luminescence in the visible region of the spectrum is 60% of that achieved in industrial phosphors. The short duration of the synthesis makes it possible to perform studies of many variants of dependencies on various factors to optimize the process.

A large amount of research remains to be done aimed at establishing the nature of processes in materials in the field of powerful radiation fluxes that ensure high synthesis efficiency, establishing the dependence of the radiative properties of ceramics on the prehistory of the starting materials (purity, fineness), optimizing the compositions of activators, synthesis modes (electron energy, power beam) and others. Establishing the possibility of using the radiation synthesis of materials based on YAG with a complex composition of activators, modifiers, and other refractory dielectric materials seems promising.

Thank you!

Reviewer 2 Report

Line 56-59. This is not precise. I understand that it is about synthesis using a high-power, high-energy electron beam. This should be worded more clearly

Line 60-61; should be formulated differently, e.g. the mixture used for the synthesis had a stoichiometric composition ........

Line 63.  of 0.2–2 wt% of the total weight. Total weight of what? Be more precise please.

Line 65-66; not clear. The „chargé” in my opinion is not the proper term in this case.

with a specified Al/Ga ratio? How this ratio was specified?

Line 67; replaced „by” and not „with”

Line 68-69; should be rewritten;  The synthesis was carried out by direct irradiation of the prepared mixture with electron beam having energy etc.....

Do not use the word “charge” to mean the content of the crucible

Additionally, please expand all abbreviations

Line 78-80. You use the word flux or flow, is that correct? Maybe use the word "beam"?

Do not use the word “charge” to mean the content of the crucible

Line 82-84 A typical view of samples …….. Maybe so: A typical photos….. In my opinion, all these sentences should be worded differently. A native speaker is needed here.

Fig.1. Are these photos taken with an optical microscope?

Fig.2. Did you measure the absorbed dose?

 Is it a post-irradiation effect?

The work requires very precise correction by a native speaker. In its current form, in my opinion, it is not suitable for publication. I have no objections to the r obtained results.

Author Response

The authors thank you for your detailed analysis and helpful comments. All comments have been corrected:

1) The sentences has been corrected for "YAG ceramic samples of various compositions in a 1.4 MeV electron flux with a power density of 15-25 kW/cm2,"

2), 3) The text corrected to: The mixture used for the synthesis of ceramics from cerium-activated yttrium-aluminum garnet (YAG:Ce) had a stoichiometric composition, consisted of 57 wt% Y2O3, 43 wt% Al2O3. Cerium oxide Ce2O3 was added in an amount of 0.2–2 wt% of the total charge weight.

4), 5) The text has been corrected to: “The charge for the synthesis of cerium-activated yttrium-aluminum-gallium garnet (YAGG:Ce) ceramics consisted of oxides Y2O3, Al2O3, Ga2O3 with 1 wt% Ce2O3 with different Al/Ga ratios. During synthesis, Ga enters the lattice by substituting Al ions. The ratio of Al/G ions varied from 0 to 1. YAGG:Ce ceramics was also synthesized with the replacement of yttrium with 6gadolinium Gd up to 15%.»

6), 7) The text has been corrected to: “The synthesis was carried out by direct irradiation of the prepared mixture with an electron beam with an energy of 1.4 MeV and a power density P = 15-25 kW/cm2 of the ELV-6 accelerator of the INP named after. Budker SB RAS.

8) An insert has been made in the text with an explanation of the caption to Fig. 1. "obtained using an optical microscope"

9) The value of the absorbed dose is given in line 82. “At P=20 kW/cm2, with the indicated beam sizes, scanning modes, charge layer thickness, the absorbed energy by the charge was 20 kJ/cm2 or 15 kJ/g.”

We submit the article for translation and proofreading to professional MDPI translators. We hope for a quality translation.

Thank you!

Reviewer 3 Report

There are some observations:

1. As the (YAG: Ce) is a well-known phosphor the luminescence section 2.3. has to be improved and discussed by using references of YAG:Ce phosphors or other Ce-doped phosphors:

- The luminescence and excitation peaks from the Figure 4a need proper assignment

- There is luminescence  lifetime and not a “relaxation time” and the value of 60-65 ns has to be discussed by using references of YAG:Ce phosphors.

2. Few comments/statements about the relevance of the novel “growth” method and expected impact in the field would be welcome, at the end of the discussion section

Author Response

The authors thank you for your detailed analysis and helpful comments. All comments have been corrected.

Added in section 2.3: 

1) Briefly additional information is provided for comparison with other phosphors, and the reasons for the spread of peak position values are also explained here: The positions of the excitation and luminescence band maxima may differ depending on the prehistory of the starting materials used for the synthesis and the synthesis modes. For example, depending on the synthesis modes, YAM and YAP crystal phases are detected in addition to the basic YAG in the material [21,22]. Introduction of Gd, Ga ions as a modifier into the lattice leads to a shift of luminescence bands to the long and short wavelength regions of the spectrum [23,24]. The probable cause of the shift in the position of the bands is a change in the lattice parameter and, accordingly, a change in the mutual arrangement of levels in the activator ion.

2) Corrected the word "relaxation time" to "luminescence decay time"

3) the section "Conclusion" has been added to the manuscript.

4) We submit the article for translation and proofreading to professional MDPI translators. We hope for a quality translation.

Added "Conclusion: 

The presented results demonstrate the possibility of a new method for the synthesis of luminescent YAG:Ce ceramics by direct irradiation with electrons with an energy of 1.4 MeV and a flux density of 15–23 kW/cm2 of a prepared mixture of stoichiometric composition of yttrium and aluminum oxides. The synthesis is realized in a time less than 1 s without the addition of substances facilitating the synthesis, during the synthesis there is an effective mixing of the initial powder Y2O3, Al2O3, Ga2O3 Ce2O3, an effective exchange of elements between the particles of the mixture powders. Radiation fusion is implemented for the first time. The resulting ceramics have qualitative radiative characteristics (luminescence and excitation spectra) quite similar to those known for YAG:Ce phosphors. The conversion efficiency of UV radiation with λ=450 nm into luminescence in the visible region of the spectrum is 60% of that achieved in industrial phosphors. The short duration of the synthesis makes it possible to perform studies of many variants of dependencies on various factors to optimize the process.

A large amount of research remains to be done aimed at establishing the nature of processes in materials in the field of powerful radiation fluxes that ensure high synthesis efficiency, establishing the dependence of the radiative properties of ceramics on the prehistory of the starting materials (purity, fineness), optimizing the compositions of activators, synthesis modes (electron energy, power beam) and others. Establishing the possibility of using the radiation synthesis of materials based on YAG with a complex composition of activators, modifiers, and other refractory dielectric materials seems promising.

Thank you!

Round 2

Reviewer 1 Report

The authors revised their manuscript by considering all the comments and suggestions from the manuscript. The authors have made a conclusion on the manuscript and write it as Section 5 for clarity and completeness of the manuscript. The format and all minor errors have been revised. Now the revised version of the manuscript is adequately good and meets the publication standard. Therefore, I recommend the revised version of the manuscript for publication.

Author Response

The authors thank you for your careful and laborious work. Your recommendations regarding the translation of the article will be implemented.

All reviewers draw attention to the insufficient quality of the translation. We agree with the remark. We wanted to submit the article for translation and proofreading to professional MDPI translators. We hope for a quality translation.

Reviewer 2 Report

After introducing these few corrections, the work is accepted for publication.

Line 51: instead „fluxes”  should be rather „beams”

Line 56: exposing the charge to flows of high-energy electrons - is the word “charge” correct? and further in the text

Fig.2 could be clearer, especially the legend

Line 97: what means;” increasing-decreasing Gaussian electron beam”. It's probably Gaussian  shaped electron beam

Fig.3 is very difficult to analyze. It needs to be more readable.

Tab.1. How Rwp[%] was calculated, please explain

Eq.1. How did Eq. (1) come about? Maybe a reference is needed?

Fig.4b. Not all parameters e.g. k1,2, t1,2 are explained

Author Response

Thank you for your comments and your time

Line 51: Replaced "fluxes" with "beams" throughout text

Line 56: The term "charge" has been changed to "mixture"

Fig. 2. With legend to fig. 2 has a problem. The calculation program does not provide for the possibility of correcting the lines denoting the boundaries of the absorbed energy density distribution. Therefore, for clarity, the central part is highlighted in color - a red oval.

Line 97. Correction done.

Fig. 3 The picture quality is good. You just need to increase the Fig. as much as possible

Table 1. Rwp(%) is the designation accepted by structural analysts for the maximum deviation from a reliable value. This value is included in modern programs for analyzing XRD results. You can remove this column. The value of Rwp(%) is given only in recent years and not always.

Equation 1 is deciphered. All formulas are standard from thermal physics.

Fig. 4b. is Corrected